# The Mystery of Certain *Lactobacillus acidophilus* Strains in the Treatment of Gastrointestinal Symptoms of COVID-19: A Review

**DOI:** 10.3390/microorganisms13040944

**Published:** 2025-04-19

**Authors:** Belén Bertola, Amparo Cotolí-Crespo, Nadia San Onofre, Jose M. Soriano

**Affiliations:** 1Food & Health Lab, Institute of Materials Science, University of Valencia, 46980 Paterna, Spain; bertola@alumni.uv.es; 2Faculty of Psychology and Speech Therapy, University of Valencia, 46010 Valencia, Spain; amparo.cotoli@uv.es; 3NUTRALiSS Research Group, Faculty of Health Sciences, Universitat Oberta de Catalunya, Rambla del Poblenou 156, 08018 Barcelona, Spain; nsan_onofre@uoc.edu; 4Joint Research Unit on Endocrinology, Nutrition and Clinical Dietetics, University of Valencia-Health Research Institute La Fe, 46026 Valencia, Spain

**Keywords:** *Lactobacillus acidophilus*, probiotics, COVID-19, gastrointestinal symptoms, gut microbiota, inflammation, gut–lung axis

## Abstract

COVID-19 presents a wide range of symptoms, including gastrointestinal manifestations such as diarrhea, nausea, and abdominal pain. *Lactobacillus acidophilus* has been proposed as a potential adjunct therapy to alleviate these symptoms due to its probiotic properties, which help restore gut microbiota balance and modulate immune responses. This review systematically analyzed studies assessing the effects of *L. acidophilus* in COVID-19 patients with gastrointestinal symptoms. The literature search was conducted through PubMed and the WHO COVID-19 database using keywords such as “*Lactobacillus acidophilus*”, “COVID-19”, “gastrointestinal symptoms”, and “inflammation markers”. The search covered studies published until February 2025. Inclusion criteria: observational and clinical trials with *L. acidophilus* for symptom relief. Exclusion: animal studies and non-ethical approvals. The findings suggest that *L. acidophilus* supplementation may contribute to faster resolution of diarrhea, improved gut microbiota balance, and reduced inflammatory markers. However, some studies have found no significant impact on hospitalization rates or disease progression. The probiotic’s mechanisms of action appear to involve microbiota modulation, intestinal barrier reinforcement, and anti-inflammatory effects rather than direct viral inhibition in COVID-19 after progression. Some *L. acidophilus* strains show promise, and clinical validation should follow careful preclinical studies (in vitro, cell lines, and animal models), especially in vulnerable populations such as immunocompromised individuals. Understanding the gut–lung axis and its role in immune response regulation, together with the need for a thorough characterization of the specific strains, including biochemical, genomic, and functional properties, before testing in humans, may provide deeper insights into the therapeutic potential of probiotics in viral infections.

## 1. Introduction

In December 2019, a new disease caused by a coronavirus, later named COVID-19, was first identified in Wuhan, China [1]. This disease was linked to a β-coronavirus similar to the strains responsible for severe acute respiratory syndrome (SARS-CoV) [2] and Middle East respiratory syndrome (MERS-CoV) [3]. The virus was officially named SARS-CoV-2 [4]. On 11 March 2020, the World Health Organization (WHO) declared COVID-19 a global pandemic [5]. The clinical manifestations of SARS-CoV-2 infection vary widely, from asymptomatic to mild/moderate, severe, and critical forms, with corresponding mortality rates of 0%, 15%, and 50%, respectively. Mild cases typically include symptoms such as dry cough, sore throat, and fever, while more severe cases progress to pneumonia and difficulty breathing. In critical cases, respiratory failure, septic shock, and multiple organ dysfunction can occur [6,7,8]. Additionally, gastrointestinal issues, including vomiting, abdominal pain, and diarrhea, have been frequently observed [9,10], possibly linked to the involvement of angiotensin-converting enzyme 2 (ACE2), which serves as the receptor facilitating SARS-CoV-2 entry into host cells [11]. ACE2 expression is notably high in various organs, including the gastrointestinal tract, lungs, heart, and kidneys [12]. Specifically, diarrhea has been observed both in the early stages of infection [13] and during ongoing infection [14], and may even persist even after recovery.

Probiotics have been proposed as a potential therapeutic approach for preventing antibiotic-associated diarrhea [15], primarily by enhancing immune responses, competing for essential nutrients, preventing pathogen adhesion to epithelial and mucosal surfaces, reducing epithelial invasion, and producing antimicrobial compounds [16,17]. The first global references discussing the use of probiotics in the context of COVID-19 were published in the British Medical Journal [18,19]. However, scientific evidence regarding the role of probiotics in COVID-19 remains fragmented across various publications, including those that clearly define specific probiotic types [20] and others that lack detailed information (grey literature) [21], as well as clinical practice guidelines [22]. In one reported case, Ji et al. [23] mentioned that “oral probiotics were given to a boy with COVID-19 symptoms, and his symptoms disappeared after two days of treatment”. Following an inquiry to the author by one of the authors of this manuscript (J.M. Soriano, personal communication, 7 April 2020), Ji clarified that the probiotic used was *Lactobacillus acidophilus* (Li-Na, personal communication, 8 April 2020).

In fact, the genus *Lactobacillus* represents a large and diverse group of lactic acid bacteria (LAB) [24], which are widely used in the food [25], pharmaceutical [26], and agricultural [27] industries due to their probiotic properties. *L. acidophilus* exhibits stronger acid tolerance, allowing it to survive harsh gastric conditions and reach the small intestine, where it contributes to microbiota balance, immune modulation, and pathogen inhibition [28]. Based on Ji’s personal communication and manuscript [23], we sought to investigate whether this microorganism could be used to address COVID-19-related gastrointestinal symptoms. To this end, a review was conducted to assess whether *L. acidophilus* could potentially serve as a therapeutic option to alleviate gastrointestinal symptoms in COVID-19 patients.

## 2. Materials and Methods

### 2.1. Focused Question

Can *L. acidophilus* contribute to the mitigation of gastrointestinal and systemic symptoms in COVID-19 patients?

### 2.2. Eligibility Criteria

First, we analyzed studies according to the following inclusion criteria:

Type of studies. Observational studies and clinical trials.

Type of participants. COVID-19 patients (reflecting information about age, sex, and other demographic factors) who received *L. acidophilus* as an intervention.

Type of interventions. Studies that evaluated the effects of *L. acidophilus* on COVID-19 patients.

Outcome type. Symptomatic relief, modulation of gut microbiota, reduction in inflammation markers, and overall patient recovery.

In the second phase, we included only studies that met all the inclusion criteria while applying the following exclusion criteria: (i) studies that did not report at least one of the selected outcome parameters, (ii) studies involving participants with severe underlying systemic diseases that could have influenced outcomes, (iii) in vitro or animal-based studies, and (iv) studies conducted without ethical approval.

### 2.3. Search Strategy

This systematic narrative review adheres to the guidelines set by the Preferred Reporting Items for Systematic Reviews and Meta-Analyses (PRISMA) [29] and was conducted using the WHO COVID-19 publications database alongside PubMed. The search methodology was designed based on the Population, Intervention, Comparator, Outcome (PICO) framework [30]. We included studies published in English and other languages, which were translated using Google Translate (Google, Inc., Mountain View, CA, USA). For manuscripts written in Asian languages, assistance was provided by the Confucius Institute at the University of Valencia prior to reviewing the full texts. Two separate teams of reviewers, skilled in medical assessments and research methods, independently evaluated studies for eligibility, relevance, and data collection, following standardized institutional procedures.

### 2.4. Research

We carried out a literature search including three main searches based on coronavirus (as a causative agent or disease) and probiotics: “novel coronavirus” OR “2019 Novel Coronavirus” OR “2019-nCoV” OR “SARS-CoV-2” OR “COVID-19” OR “coronavirus disease 2019” AND “probiotics” OR “intestinal microecological modulator” (according to Wei et al. [21]) OR “probiotics” OR “*Lactobacillus*” OR using the new names for the probiotic species of *Lactobacillus* [28], such as “*Lacticaseibacillus*” AND “gastrointestinal symptoms” OR “diarrhea” OR “nausea” OR “vomiting” OR “abdominal pain” OR “gut inflammation” OR “gastrointestinal distress” OR “irritable bowel” OR “intestinal dysbiosis” OR “microbiota imbalance” OR “gut–lung axis” OR “COVID-19 gastrointestinal symptoms” OR “gastrointestinal COVID-19 complications”. The search date was until 2 February 2025. Based on information from the National Heart, Lung and Blood Institute [31], the validity of each included study was assessed using nine items, to which the answer was affirmative (+), negative (–), or other, including “cannot determine”, “not applicable”, and “not reported”, which were considered unclear (?) answers. We have classified the studies using a rating of good (7–9), fair (4–6), or poor (≤3) for each individual study.

## 3. Results and Discussion

### 3.1. Selection of Results

The PRISMA flow diagram for this narrative review is presented in Figure 1. From a total of 75,120 records identified through database searches, 5043 duplicate records were removed, leaving 70,077 for screening. After an initial assessment, 69,149 records were excluded because they involved animal studies or lacked full-text availability. This left 928 studies eligible for screening. Of these, 857 studies were excluded because they did not investigate *L. acidophilus* as a probiotic. The remaining 71 full-text articles were assessed for eligibility. During this stage, 64 articles were further excluded because they did not focus on the treatment of gastrointestinal symptoms with *L. acidophilus*, together with articles excluded for being bibliographic reviews. After applying the eligibility criteria, a total of seven observational studies were included in the qualitative synthesis, but several completed trials, including NCT05474144 [32] and NCT04420676 [33], were not included because the results were not publicly available, limiting quantitative analysis. These studies were analyzed to determine the effects of *L. acidophilus* on COVID-19 symptoms, focusing on gastrointestinal outcomes.

The methodological quality of the selected studies was assessed using the National Heart, Lung, and Blood Institute (NHLBI) criteria [29], as demonstrated in Table 1.

The quality rating is presented in Table 1. Among the seven studies included in the qualitative synthesis, six studies were rated as good quality, and one study was considered fair quality. Most high-quality studies provided a clear research question, well-defined intervention protocols, and robust statistical analyses. Conversely, studies rated as fair had limitations, such as unclear descriptions of intervention methods, inconsistent reporting of outcomes, or inadequate statistical approaches.

### 3.2. Selected Studies

The study by Ji et al. [23] demonstrated the use of *L. acidophilus* in managing gastrointestinal symptoms in COVID-19 patients (15- and 9-year-old boys). In one documented case, a boy presented mild diarrhea as the only symptom of COVID-19, without fever or respiratory distress. The child was administered an oral probiotic containing *L. acidophilus* and the diarrhea resolved completely within two days. This case highlights the potential benefit of probiotics in alleviating gastrointestinal manifestations of COVID-19, particularly the kind of symptoms that the patient showed. While the study does not provide extensive details on probiotic treatment protocols, the rapid symptom resolution following probiotic administration suggests a clinically relevant effect. Horowitz et al. [34] studied the use of probiotics, including *L. acidophilus*, in the treatment of symptoms such as diarrhea due to COVID-19. In a 53-year-old male patient, *L. acidophilus* was administered as part of an immune and nutritional support regimen, alongside *Bifidobacterium* and *Saccharomyces boulardii*, in combination with an antibiotic regimen. The study suggests that this probiotic mixture was associated with improved gut health and immune function, though it does not specify direct clinical outcomes related solely to *L. acidophilus*. The probiotic supplementation was used in conjunction with zinc (40 mg/day), Vitamin C (up to 2 g, three times daily), beta-glucan (1000 mg daily), curcumin (1 g twice daily), sulforaphane glucosinolates (100 mg twice daily), N-acetyl-cysteine (NAC) (600 mg twice daily), alpha-lipoic acid (600 mg twice daily), and glutathione (500 mg capsules twice daily, increased up to 2 g pro re nata (PRN) for acute respiratory distress). These findings suggest that probiotics, including *L. acidophilus*, may contribute to gut microbiota modulation and systemic immune regulation, potentially aiding in managing COVID-19-associated gastrointestinal symptoms. The study by Feng et al. [35] highlights the gastrointestinal implications of SARS-CoV-2 infection, emphasizing that enterocytes serve as a viral reservoir due to the high expression of ACE2 receptors in the small intestine, which may contribute to COVID-19-associated diarrhea and prolonged viral shedding in feces. The study reports that probiotics, including *L. acidophilus*, were widely used in Wuhan for COVID-19 patients experiencing diarrhea, aiming to restore gut microbiota balance. However, the findings indicate that while *L. acidophilus* and *Bacillus clausii* contributed to microbial modulation, they did not significantly reduce ACE2 receptor expression in the gut, suggesting that probiotics may help alleviate inflammation and dysbiosis but may not directly block viral entry into intestinal cells.

On the other hand, Saviano et al. [36] investigated the impact of a probiotic mix, including *L. acidophilus* LA 201, *Bifidobacterium lactis* LA 304, and *Lactobacillus salivarius* LA 302, on gastrointestinal inflammation in COVID-19 patients. The results showed that probiotic supplementation led to a significant reduction in fecal calprotectin levels, a key marker of gut inflammation, with a 35% decrease in the probiotic group compared to only 16% in the control group. Additionally, C-reactive protein (CRP), a systemic inflammatory marker, decreased by 72.7% in the probiotic-treated patients, whereas the control group exhibited a reduction of 62%. The study also observed a trend toward a lower need for oxygen support in the probiotic group, although this result was not statistically significant. These findings suggest that *L. acidophilus*, in combination with other probiotics, may help reduce gut and systemic inflammation in COVID-19 patients, highlighting its potential role as an adjunct therapy in managing gastrointestinal symptoms associated with the disease. Gutiérrez-Castrellón et al. [37] evaluated the effects of a daily oral capsule containing 2 × 10^9^ CFU of a probiotic formula, which contained *Lactiplantibacillus plantarum* strains KABP022, KABP023, KABP033 (which are part of the Lactobacillaceae family), and *Pediococcus acidilactici* strain KABP021. Furthermore, they were instructed to use it for 30 days, before breakfast, instead of from day 1 to day 30. The probiotic product was stored at room temperature and monitored for its microbial quality throughout the study. During these visits, various assessments were made, including the severity of COVID-19 symptoms, chest X-rays, and blood and fecal sample collections. The participants were also contacted by phone on several days for outpatient follow-up. The study design adhered to ethical protocols, and the product was placebo-controlled. It was blinded, randomized, and followed CONSORT guidelines. Only acetaminophen was permitted as co-medication. Patients receiving supplementation experienced a significant reduction in diarrhea, nausea, abdominal pain, and loose stools compared to those in the placebo group. The study also noted that probiotic-treated individuals had faster overall symptom resolution, suggesting that those probiotics contributed to restoring gut homeostasis and alleviating inflammation. Furthermore, the probiotic group showed improved gut microbiota composition, with increased levels of beneficial bacteria and a decrease in pathogenic microbial species commonly associated with dysbiosis in COVID-19 patients. These findings indicate that the dose may serve as an adjunctive therapy to counteract gastrointestinal disturbances in COVID-19, helping to regulate microbial balance and reduce gut inflammation. Horvath et al. [38] analyzed the use of probiotics for gastrointestinal symptoms during COVID-19. It reports on a randomized, double-blind, placebo-controlled study where patients with mild COVID-19 in home quarantine were given a multispecies probiotic, including *L. acidophilus* W37 and W55, among other strains (*Bifidobacterium bifidum* W23, *Bifidobacterium lactis* W51, *Enterococcus faecium* W54, *Lactocaseibacillus paracasei* W20, *Lactoplantibacillus plantarum* W1/W62, and *Lactocaseibacillus rhamnosus* W71). The study aimed to assess changes in the microbiome and gastrointestinal symptoms. The findings indicated that COVID-19 patients experienced significant alterations in their gut microbiome, including a reduction in beneficial bacteria such as *Christensenellaceae* and *Ruminococcaceae* and an increase in *Bacteroidetes*. Although probiotic treatment modulated microbiome diversity and introduced *Enterococcus faecium* W54, no significant effect was observed on COVID-19-related symptoms, including diarrhea or stool consistency. While probiotics did not lead to a measurable improvement in gastrointestinal symptoms during the study period, the microbiome modulation observed suggests a potential long-term benefit. However, *L. acidophilus* was only detected in a small number of samples, indicating that its colonization may be inconsistent or require longer administration periods for a measurable effect. These authors emphasized that the beneficial effects were more associated with inflammation modulation rather than a direct impact on ACE2 receptor expression. Finally, Hassan et al. [39] reflected the effects of *L. acidophilus* and colchicine on the symptoms, duration, and disease progression of mild and moderate COVID-19 cases in a randomized controlled trial. The study included 150 patients, divided into three groups: one receiving colchicine, one receiving *L. acidophilus*, and one as a control group. The findings revealed that gastrointestinal symptoms, including diarrhea, nausea, and vomiting, were commonly reported among COVID-19 patients, and gut microbiota alterations persisted even after viral clearance. While probiotics, including *L. acidophilus*, were administered to improve gut health and modulate the immune response, the study found no statistically significant difference in symptom improvement, disease duration, or hospitalization rates between the probiotic group and the control group. These results suggest that while *L. acidophilus* may support gut microbiota balance, it did not significantly alter clinical outcomes in mild and moderate COVID-19 patients in this particular study.

### 3.3. Mechanism of Action of L. acidophilus in Counteracting Gastrointestinal Effects

*L. acidophilus* exerts its beneficial effects on gastrointestinal health through multiple mechanisms, including modulation of gut microbiota, reinforcement of the intestinal barrier, immunomodulation, interaction with the gut–brain axis, antiviral properties, and regulation of ACE2 expression. These mechanisms contribute to its ability to counteract gastrointestinal disturbances, including those associated with infections, inflammatory diseases, and functional disorders. The first mechanism through which *L. acidophilus* provides gastrointestinal benefits is by modulating gut microbiota and competing with pathogenic bacteria. This probiotic has the ability to colonize the intestinal mucosa and form a protective biofilm, which prevents the adhesion and overgrowth of harmful bacteria such as *Salmonella typhimurium*, *Escherichia coli*, and *Clostridium difficile* [40,41]. It produces bacteriocins, antimicrobial peptides that inhibit the growth of pathogens, and organic acids such as lactic acid and acetic acid, which lower the gut pH and create an inhospitable environment for many harmful microorganisms [42]. By maintaining a balanced microbiota composition, *L. acidophilus* helps prevent dysbiosis, a condition associated with several gastrointestinal disorders, including irritable bowel syndrome (IBS) and inflammatory bowel disease (IBD) [43]. Studies have shown that supplementation with *L. acidophilus* can lead to an increase in beneficial bacteria such as *Bifidobacterium* and a decrease in opportunistic pathogens, thereby improving gut health and reducing symptoms such as bloating, diarrhea, and constipation [44].

Another crucial function of *L. acidophilus* is its role in strengthening the intestinal barrier. The intestinal epithelium serves as the first line of defense against pathogens and toxins, and its integrity is essential for maintaining gut health [45]. *L. acidophilus* enhances the production of tight junction proteins, including occludin, claudin-1, and zonula occludens-1 (ZO-1), which are responsible for sealing the spaces between epithelial cells and preventing the translocation of harmful bacteria and toxins into the bloodstream [46]. Additionally, this probiotic stimulates the secretion of mucins (MUC2, MUC3), which reinforce the mucus layer that lines the gut, acting as a physical barrier against microbial invasion [47]. Studies have demonstrated that *L. acidophilus* supplementation can restore intestinal barrier integrity in conditions such as leaky gut syndrome [48], Crohn’s disease [49], and ulcerative colitis [50], thereby reducing inflammation and improving overall gut function [51].

*L. acidophilus* also exerts significant anti-inflammatory and immunomodulatory effects, which play a crucial role in maintaining gastrointestinal homeostasis [52]. This probiotic modulates the immune system by promoting the production of anti-inflammatory cytokines such as interleukin-10 (IL-10) [53] while reducing the levels of pro-inflammatory cytokines such as tumor necrosis factor-alpha (TNF-α), interleukin-6 (IL-6), and interleukin-8 (IL-8) [54]. Additionally, *L. acidophilus* enhances the activity of regulatory T cells (Tregs), which help in maintaining immune tolerance and preventing excessive inflammatory responses [55]. This immunomodulatory effect is particularly beneficial in conditions such as IBD and IBS, where an overactive immune response contributes to chronic inflammation and tissue damage [56]. By modulating immune function, *L. acidophilus* not only helps in reducing gut inflammation but also supports systemic immune health [57].

The interaction of *L. acidophilus* with the gut–brain axis represents another important mechanism by which it alleviates gastrointestinal symptoms [58]. The gut–brain axis is a bidirectional communication network between the central nervous system and the enteric nervous system, influenced by gut microbiota and microbial metabolites [59]. *L. acidophilus* has been shown to modulate the production of neurotransmitters such as gamma-aminobutyric acid (GABA) and serotonin, which play essential roles in regulating gut motility, visceral sensitivity [60], and mood [61]. Research indicates that supplementation with *L. acidophilus* can help in reducing stress-induced gastrointestinal symptoms, such as diarrhea, abdominal pain, and bloating, by modulating vagus nerve activity and reducing the perception of gut discomfort [13]. Additionally, clinical studies have demonstrated that probiotics, including *L. acidophilus*, may have potential therapeutic applications in conditions such as anxiety, depression, and functional gastrointestinal disorders, where gut–brain interactions are disrupted [62].

Another important mechanism through which *L. acidophilus* provides gastrointestinal benefits is its antiviral properties and protective effects against enteric infections [63]. This probiotic produces antimicrobial metabolites, including hydrogen peroxide [64], short-chain fatty acids (SCFAs) [65], and bacteriocins [66], which exhibit antiviral activity against enteric viruses such as rotavirus [67] and norovirus [68]. Additionally, *L. acidophilus* enhances the secretion of mucins [69] and immunoglobulin A (IgA) [70], which contribute to the neutralization and clearance of viral pathogens from the gut. Studies have shown that *L. acidophilus* may reduce the severity and duration of viral gastroenteritis in some COVID-19 patients by improving immune response and gut barrier function [71,72].

On the other hand, SARS-CoV-2 has been suggested to exhibit bacteriophage-like behavior, interacting with the gut microbiota and possibly altering the microbial composition, leading to gastrointestinal symptoms such as diarrhea and vomiting. This interaction is thought to occur through the virus’s use of the ACE2 receptor, which is highly expressed not only in the lungs but also in the gastrointestinal tract, making the gut a key site for viral entry. Studies indicate that the virus could influence gut microbiota by modifying bacterial populations, similar to the mechanisms of bacteriophages, which infect and alter bacterial behavior. This has important implications for understanding how the virus contributes to gut dysbiosis and inflammation and highlights the potential role of probiotics in modulating these effects, potentially reducing viral load and improving gastrointestinal health. A better understanding of these interactions can guide therapeutic strategies for managing COVID-19 symptoms related to the gastrointestinal system [73].

Finally, the regulation of ACE2 expression in the gut represents a novel area of research regarding the potential implications of *L. acidophilus* in COVID-19-associated gastrointestinal symptoms [74]. ACE2 is a key receptor for SARS-CoV-2 entry into host cells and is highly expressed in enterocytes of the small intestine [75]. The challenge then is finding potential probiotic strains able to significantly reduce ACE2 receptor expression in a gut cell line. Some studies suggest that probiotics, including *L. acidophilus*, may influence ACE2 expression and modulate gut inflammation during viral infections [76,77,78].

## 4. Conclusions

This review highlights the potential of some *L. acidophilus* strains and their combinations with other bacteria in alleviating gastrointestinal symptoms in COVID-19 patients. Evidence suggests it may help modulate gut microbiota, regulate immune responses, reduce inflammation, and enhance the production of tight junction proteins, reinforcing the intestinal barrier. The primary mechanism appears to involve gut inflammation and dysbiosis modulation rather than direct viral inhibition. However, some probiotic strains, including *L. acidophilus*, have also been shown to influence the expression of ACE2 receptors. ACE2, which is the entry point for SARS-CoV-2 into host cells, is highly expressed in the intestinal epithelial cells. By modulating the expression of ACE2 receptors, *L. acidophilus* may indirectly affect the host’s susceptibility to the virus. This effect could help reduce viral entry and subsequent inflammation, further supporting its role in managing COVID-19-related gastrointestinal symptoms. Therefore, *L. acidophilus* may provide a dual benefit by enhancing the intestinal barrier through tight junction modulation and potentially influencing ACE2 expression, which could complement the immune response against viral infections. However, further rigorous investigation is required to establish optimal dosages, type of strain, treatment duration, and the specific patient populations that may derive the most benefit, in addition to elucidating the role of the gut–lung axis in immune response regulation.

## Figures and Tables

**Figure 1 microorganisms-13-00944-f001:**
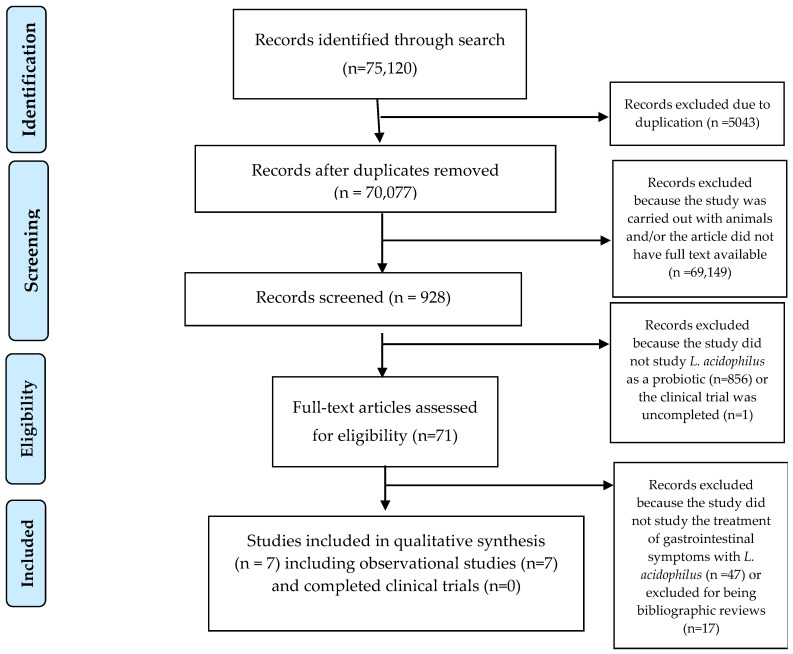
PRISMA flow diagram for studies retrieved through the searching and selection process.

**Table 1 microorganisms-13-00944-t001:** Methodological quality assessment of studies selected for this review, according to the criteria of the National Heart, Lung and Blood Institute [31].

Reference/ Article	1 ^a^	2 ^a^	3 ^a^	4 ^a^	5 ^a^	6 ^a^	7 ^a^	8 ^a^	9 ^a^	Quality Rating ^b^
Ji et al. [23]	+	+	?	+	+	+	?	+	+	7
Horowitz et al. [34]	+	+	+	+	+	+	−	+	+	8
Feng et al. [35]	+	+	−	+	+	−	−	−	−	4
Saviano et al. [36]	+	+	+	+	+	+	+	+	+	9
Gutiérrez et al. [37]	+	+	+	+	+	+	+	+	+	9
Horvarth et al. [38]	+	+	+	+	+	+	+	+	+	9
Hassan et al. [39]	+	+	+	+	+	+	−	−	+	7

Affirmative (+), negative (−), or other answers, including “cannot be determined”, “not applicable”, and “not informed”, were considered unclear (?) answers. ^a^ The National Heart, Lung, and Blood Institute articles [31] were: 1 = Was the question or objective of the study clearly stated?; 2 = Was the study population clearly and completely described, including a case definition?; 3 = Were the cases consecutive?; 4 = Were the subjects comparable?; 5 = Was the intervention clearly described?; 6 = Were outcome measures clearly defined, valid, reliable, and consistently implemented across study participants?; 7 = Was the duration of follow-up adequate?; 8 = Were the statistical methods well described?; and 9 = Were the results well described? ^b^ Quality rating [31] was good (7–9), fair (4–6), or poor (≤3).

## Data Availability

The data presented in this study are available on request from the corresponding author. The data are not publicly available due to privacy reason.

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
