# Peer review of "The Mystery of Certain Lactobacillus acidophilus Strains in the Treatment of Gastrointestinal Symptoms of COVID-19: A Review"

_microorganisms, 2025, doi:10.3390/microorganisms13040944_

Round 1

Reviewer 1 Report

Comments and Suggestions for Authors

The authors present a succinct review of the potential of L. acidophilus in alleviating gastrointestinal symptoms in COVID-19 patients. The data reported suggest that this type of probiotic may help modulate gut microbiota, regulate immune responses, and reduce inflammation. The possible mechanism through which L. acidophilus provides gastrointestinal benefits is based on its antiviral properties, which are due to its metabolites such as hydrogen peroxide, short-chain fatty acids, and bacteriocins, which exhibit antiviral activity against enteric viruses such as rotavirus and norovirus. Moreover, the probiotic L. acidophilus enhances the secretion of mucins and immunoglobulin A (IgA), which contribute to the neutralisation and clearance of viral pathogens from the gut. The manuscript is well-written, concise, and interesting, and reveals the importance of probiotics in maintaining digestive health in cases of severe illness due to coronavirus. For all these reasons, I recommend the manuscript for publication.

Author Response

Reviewer’s comment: The authors present a succinct review of the potential of L. acidophilus in alleviating gastrointestinal symptoms in COVID-19 patients. The data reported suggest that this type of probiotic may help modulate gut microbiota, regulate immune responses, and reduce inflammation. The possible mechanism through which L. acidophilus provides gastrointestinal benefits is based on its antiviral properties, which are due to its metabolites such as hydrogen peroxide, short-chain fatty acids, and bacteriocins, which exhibit antiviral activity against enteric viruses such as rotavirus and norovirus. Moreover, the probiotic L. acidophilus enhances the secretion of mucins and immunoglobulin A (IgA), which contribute to the neutralisation and clearance of viral pathogens from the gut. The manuscript is well-written, concise, and interesting, and reveals the importance of probiotics in maintaining digestive health in cases of severe illness due to coronavirus. For all these reasons, I recommend the manuscript for publication.

Author’s comment: Thank you for your comment.

Reviewer 2 Report

Comments and Suggestions for Authors

The authors underline the importance of L. acidophilus in the treatment of gastrointestinal symptoms during COVID-19 infection. The review is based on many studies that underline this aspect. The prisma protocol is clear and the authors have discussed the field but have forgotten to discuss the major studies that show the bacteriophage behaviour of SARS-CoV-2. It is important discuss this aspect in the introduction and in the discussion.

It is important to be clear also for the abbreviations and the conclusion would be  more extensive.

Author Response

Reviewer’s comment: The authors underline the importance of L. acidophilus in the treatment of gastrointestinal symptoms during COVID-19 infection. The review is based on many studies that underline this aspect. The prisma protocol is clear and the authors have discussed the field but have forgotten to discuss the major studies that show the bacteriophage behaviour of SARS-CoV-2. It is important discuss this aspect in the introduction and in the discussion.

Author’s comment:  It has added in the manuscript

Reviewer’s comment: It is important to be clear also for the abbreviations and the conclusion would be  more extensive.

Author’s comment: According to your comment, it has been added.

Reviewer 3 Report

Comments and Suggestions for Authors

Line 17:

Use: which help to restore

Line 19: It is necessary to provide more details about how the information search was conducted, for example, keywords used, time, and inclusion and exclusion criteria.

Line 23-26: at the end of the statement include “in COVID-19” or the disease you are talking about. Decide if you put “in COVID-19 after progression or use “than direct COVID-19 inhibition.

Line 26: I do not agree with L. acidophilus shows promise in the management of gastrointestinal symptoms associated with COVID-19 because the effect of probiotics depends on the genera and specie of a strain, and it is not possible generalize over all L. acidophilus. I would prefer that you include the name of the specific strains of L. acidophilus that show this effect. I also recommend that modify the title since it seems that all L. acidophilus have this effect at it is not true. In addition, I consider that it is necessary to establish in which sceneries is possible to administer this probiotic and in which not.

Line 27-29: It is important that the effect be first documented in in vitro studies, cell lines, and animal models before being tested in humans under certain health conditions, since regardless of whether it is a probiotic, for immunocompromised individuals it could mean the loss of life. Considering this comment pleas redraft “however, further 27 large-scale clinical trials are required to validate its therapeutic efficacy.”

Line 28-28: I agree in that is necessary to have a better understood of gut-lung-axis …….  However, which is first the most important is a deep characterization of the potential probiotic strains, including biochemical, genomic, and functional characterizations in vitro and in vivo before probing them in humans.

Line 80: I strongly think that one of the criteria to conduct the search is the age of the patient since as you found in line 61 the possibly effects were observed in a boy, thus is clear that the immune response of a boy is strongly different to the response of an older people. Other criteria that is important to consider is the sex, since women and men respond in a different way since hormones are involved in. Or in other words, the population in which was observed the effect have to be described somewhere in the paper.

Figure 1 use L. acidophillus in the third box

Line 150-151 and 152. The information about quality rating es defined in duplicate, please decide which lines are better for define this information.

Line 160: Pediatric is a very broad term, please define the characteristics of this pediatric population

Line 162: define routine treatment since the therapy using probiotics is not always compatible with all kinds of drugs.

Lines 165 include “with the kind of symptoms that the patient showed” please do not generalize.

Lines 168: In the treatment of COVID-19 or in the treatment of the symptoms such as diarrhea due to COVID-19????

Lines 167-180 I agree with the findings of Horowitz , however to this paper is not possible to consider mixes of microorganism because the effect have not totally attributed to the strain in question (L. acidophilus).

Line 184: please include the specific strain.

Line 186-189: Regarding this information the challenge then is find potential probiotic strains able to significantly reduce ACE2 receptor expression in a gut cell line. Please take this into account in the discussion.

Lines 193 to 202 : I agree with this information but The question is about the effect of L. acidophilus o about mixes of probiotics in which L. acidophilus is part of?

Line 206 please describe as possible it is the study, doses, time of doses, via of administration type of “pharmaceutic form” and other details.

Line 207 . please specify the strain, do not generalize.

Line 210 : include examples of these beneficial bacteria  and pathogenic microbial species associated with dysbiosis in COVID-19 patients.

Line 211 please specify the strain, do not generalize.

Lines 213-214: which findings of the study support this line? “and possibly shorten the duration  of viral symptoms linked to intestinal involvement.”

Line 216  please specify the strain, do not generalize.

Lines 220-221: which other strains. The same the question of this paper is about L. acidophilus o mixes in which it is included?

Lines 222-231: The information described in this paragraph empathized why it has to  use strains that previously demonstrated significantly reduce ACE2 receptor expression or that they can reduce gut and systemic inflammation in a cell line or murine model.

Line 232: please specify the strain, do not generalize.

Lune 252 please include a cite after C. difficile. Since citation number 40, do not include biofilm production. In addition, citation 40 only include Salmonella thypimurium.  This citation do not include bacteriocins, pH, environmental conditions  and antimicrobial peptides as mechanism of action of L. acidophilus  to inhibit pathogens.

Line 307 use italics to L. acidophilus

Lines 307-308 include a citation at the end of the paragraph and do not generalize using children  and adults include more details about the studied population.

Lines 316 I would say the potential of some strains of L. acidophilus that show potential in the alleviation of gastro… symptoms in  XXXX type of patients with COVID-19.

Line 318: and what about tight junctions?

Lines 319-320: , This statement was nod discussed in the paper  “while others indicate no significant impact on overall disease progression or hospitalization rates”

Line 322: and what about ACE2?

Line 322-324: Include type of strain

Final comment : I ran your article through anti-plagiarism software and it shows 54% of it.

Author Response

Reviewer’s comment: Line 17: Use: which help to restore

Author’s comment: According to your comment, we have incorporated this idea.

Reviewer’s comment: Line 19: It is necessary to provide more details about how the information search was conducted, for example, keywords used, time, and inclusion and exclusion criteria.

Author’s comment: According to your comment, we have incorporated this idea.

Reviewer’s comment: Line 23-26: at the end of the statement include “in COVID-19” or the disease you are talking about. Decide if you put “in COVID-19 after progression or use “than direct COVID-19 inhibition.

Author’s comment: According to your comment, we have incorporated this idea in these lines

Reviewer’s comment: Line 26: I do not agree with L. acidophilus shows promise in the management of gastrointestinal symptoms associated with COVID-19 because the effect of probiotics depends on the genera and specie of a strain, and it is not possible generalize over all L. acidophilus. I would prefer that you include the name of the specific strains of L. acidophilus that show this effect. I also recommend that modify the title since it seems that all L. acidophilus have this effect at it is not true. In addition, I consider that it is necessary to establish in which sceneries is possible to administer this probiotic and in which not.

Author’s comment: We completely agree with your comments, but the information you requested is not found in the selected articles regarding the necessary considerations to establish the situations in which this probiotic can be administered and where it cannot. This information has been extracted from some articles but not all. It has been specified in the results section. On the other hand, we have modified the title to indicate that it concerns some microorganisms.

Reviewer’s comment: Line 27-29: It is important that the effect be first documented in in vitro studies, cell lines, and animal models before being tested in humans under certain health conditions, since regardless of whether it is a probiotic, for immunocompromised individuals it could mean the loss of life. Considering this comment pleas redraft “however, further 27 large-scale clinical trials are required to validate its therapeutic efficacy.”

Author’s comment: : According to your comment, we have incorporated this idea in these lines.

Reviewer’s comment: Line 28-28: I agree in that is necessary to have a better understood of gut-lung-axis …….  However, which is first the most important is a deep characterization of the potential probiotic strains, including biochemical, genomic, and functional characterizations in vitro and in vivo before probing them in humans.

Author’s comment: According to your comment, we have incorporated this idea in these lines.

Reviewer’s comment: Line 80: I strongly think that one of the criteria to conduct the search is the age of the patient since as you found in line 61 the possibly effects were observed in a boy, thus is clear that the immune response of a boy is strongly different to the response of an older people. Other criteria that is important to consider is the sex, since women and men respond in a different way since hormones are involved in. Or in other words, the population in which was observed the effect have to be described somewhere in the paper.

Author’s comment: According to your comment, it has been added in the type of participant’s section.

Reviewer’s comment: Figure 1 use L. acidophillus in the third box

Author’s comment: According to your comment, it has been modified in the Figure 1 (box 3).

Reviewer’s comment: Line 150-151 and 152. The information about quality rating es defined in duplicate, please decide which lines are better for define this information.

Author’s comment: Thank you for your comment, we have re-written this paragraph to eliminate the duplicate.

Reviewer’s comment: Line 160: Pediatric is a very broad term, please define the characteristics of this pediatric population

Author’s comment: According to your comment, we have justified the age of both boys.

Reviewer’s comment: Line 162: define routine treatment since the therapy using probiotics is not always compatible with all kinds of drugs.

Author’s comment: Thank you for this idea. We have re-written this line.

Reviewer’s comment: Lines 165 include “with the kind of symptoms that the patient showed” please do not generalize.

Author’s comment: Thank you for this idea. We have added your information in this line.

Reviewer’s comment: Lines 168: In the treatment of COVID-19 or in the treatment of the symptoms such as diarrhea due to COVID-19????

Author’s comment: Sorry. Thank you for your idea. The idea adequate was in the treatment of the symptoms such as diarrhea due to COVID-19. It has been added.

Reviewer’s comment: Lines 167-180 I agree with the findings of Horowitz , however to this paper is not possible to consider mixes of microorganism because the effect have not totally attributed to the strain in question (L. acidophilus).

Author’s comment: It is clarified when authors written this idea in this paragraph… “The study suggests that this probiotic mixture was associated with improved gut health and immune function, though it does not specify direct clinical outcomes related solely to L. acidophilus”

Reviewer’s comment: Line 184: please include the specific strain.

Author’s comment: We cannot add that information because the authors of the article do not indicate it.

Reviewer’s comment: Line 186-189: Regarding this information the challenge then is find potential probiotic strains able to significantly reduce ACE2 receptor expression in a gut cell line. Please take this into account in the discussion.

Author’s comment: According to your comment, this idea has been included in the discussion section.

Reviewer’s comment: Lines 193 to 202 : I agree with this information but The question is about the effect of L. acidophilus o about mixes of probiotics in which L. acidophilus is part of?

Author’s comment: The authors of the study provide information about the probiotic mixture that includes L. acidophilus, but they did not conduct a study on the effect of L. acidophilus alone. Therefore, we believe it is more appropriate to indicate this as it is in the text, since it is the result of the authors' study.

Reviewer’s comment: Line 206 please describe as possible it is the study, doses, time of doses, via of administration type of “pharmaceutic form” and other details.

Author’s comment: According to your comment, it has been added in the manuscript.

Reviewer’s comment: Line 207 . please specify the strain, do not generalize.

Author’s comment: According to your comment, it has been added in the manuscript.

Reviewer’s comment: Line 210 : include examples of these beneficial bacteria  and pathogenic microbial species associated with dysbiosis in COVID-19 patients.

Author’s comment: It has clarified in the manuscript.

Reviewer’s comment: Line 211 please specify the strain, do not generalize.

Author’s comment: According to your comment, it has been added in the manuscript.

Reviewer’s comment: Lines 213-214: which findings of the study support this line? “and possibly shorten the duration  of viral symptoms linked to intestinal involvement.”

Author’s comment: Sorry. This part was just an assumption. We've removed it.

Reviewer’s comment: Line 216  please specify the strain, do not generalize.

Author’s comment: According to your comment, it has been added in the manuscript.

Reviewer’s comment: Lines 220-221: which other strains. The same the question of this paper is about L. acidophilus o mixes in which it is included?

Author’s comment: They studied on a multispecies probiotic mix, including Lactobacillus acidophilus among other strains. The study does not isolate L. acidophilus alone, but instead evaluates a combination of probiotic strains. Mixes of strains have been added in the manuscript.

Reviewer’s comment: Lines 222-231: The information described in this paragraph empathized why it has to  use strains that previously demonstrated significantly reduce ACE2 receptor expression or that they can reduce gut and systemic inflammation in a cell line or murine model.

Author’s comment: The paper emphasizes that the beneficial effects were more associated with inflammation modulation rather than a direct impact on ACE2 receptor expression.

Reviewer’s comment: Line 232: please specify the strain, do not generalize.

Author’s comment: Sorry, but the study does not specify the strain

Reviewer’s comment: Line 252 please include a cite after C. difficile. Since citation number 40, do not include biofilm production. In addition, citation 40 only include Salmonella thypimurium.  This citation do not include bacteriocins, pH, environmental conditions  and antimicrobial peptides as mechanism of action of L. acidophilus  to inhibit pathogens.

Author’s comment: According to your comment, we have added references to justify this paragraph.

Reviewer’s comment: Line 307 use italics to L. acidophilus

Author’s comment: According to your comment, we have changed this microorganism in italics.

Reviewer’s comment: Lines 307-308 include a citation at the end of the paragraph and do not generalize using children  and adults include more details about the studied population.

Author’s comment: References are included in this paragraph “Studies [69,70] have shown that L. acidophilus may reduce the severity and duration of viral gastroenteritis in children and adults by improving immune response and gut barrier function”. We have included these references in the end of paragraph. Furthermore, we have changed the  part of children and adults.

Reviewer’s comment: Lines 316 I would say the potential of some strains of L. acidophilus that show potential in the alleviation of gastro… symptoms in  XXXX type of patients with COVID-19.

Author’s comment: According to your comment, it has been modified.

Reviewer’s comment: Line 318: and what about tight junctions?

Author’s comment: According to your comment, we have added in the manuscript.

Reviewer’s comment: Lines 319-320: , This statement was nod discussed in the paper  “while others indicate no significant impact on overall disease progression or hospitalization rates”

Author’s comment: I agree with your statement. We have removed it from the text.

Reviewer’s comment: Line 322: and what about ACE2?

Author’s comment: According to your comment, it has been added.

Reviewer’s comment: Line 322-324: Include type of strain

Author’s comment: According to your comment, it has been added.

Reviewer’s comment: Final comment : I ran your article through anti-plagiarism software and it shows 54% of it.

Author’s comment: According to your comment, article has been reviewed to diminish the plagiarism.

Round 2

Reviewer 3 Report

Comments and Suggestions for Authors

Line 82 Ust the name of bacteria in Italics.

Line 309 use “a boy….”

Line 313 eliminate “in pediatric patients”

Line 361 eliminate “, being  each capsule provided a dose of 2×109 CFU-“

Line 358 use “ daily oral capsule containing 2×109 CFU of a probiotic formula,”…..

Line 362 : use during 30 days instead of day 1 to day 30

Line 363 you repeated the study in the same line please redraft

Line 370 do not include this capsule

Line 373 Those instead of these

Line 377: use dose instead of this capsule

Line 379 use “, and reduced….”

Line 531 some L. acidophilus and mixes with other bacteria

Line 534 eliminate these

Line 569 use furthermore instead of further

Author Response

Reviewer’s comment: Line 82 Ust the name of bacteria in Italics.

Author’s comment: According to your comment, it has been changed.

Reviewer’s comment: Line 309 use “a boy….”

Author’s comment: According to your comment, it has been changed.

Reviewer’s comment: Line 313 eliminate “in pediatric patients”

Author’s comment: According to your comment, it has been eliminated.

Reviewer’s comment: Line 361 eliminate “, being  each capsule provided a dose of 2×109 CFU-“

Author’s comment: According to your comment, it has been eliminated.

Reviewer’s comment: Line 358 use “ daily oral capsule containing 2×109 CFU of a probiotic formula,”…..

Author’s comment: According to your comment, it has been changed.

Reviewer’s comment: Line 362 : use during 30 days instead of day 1 to day 30

Author’s comment: According to your comment, it has been changed.

Reviewer’s comment: Line 363 you repeated the study in the same line please redraft

Author’s comment: According to your comment, it has been changed.

Reviewer’s comment: Line 370 do not include this capsule

Author’s comment: According to your comment, it has been changed.

Reviewer’s comment: Line 373 Those instead of these

Author’s comment: According to your comment, it has been changed.

Reviewer’s comment: Line 377: use dose instead of this capsule

Author’s comment: According to your comment, it has been changed.

Reviewer’s comment: Line 379 use “, and reduced….”

Author’s comment: According to your comment, it has been changed.

Reviewer’s comment: Line 531 some L. acidophilus and mixes with other bacteria

Author’s comment: According to your comment, it has been changed.

Reviewer’s comment: Line 534 eliminate these

Author’s comment: According to your comment, it has been eliminated.

Reviewer’s comment: Line 569 use furthermore instead of further

Author’s comment: According to your comment, it has been changed.